# Face and Content Validity of the Pictorial Scale of Perceived Water Competence in Young Children

**DOI:** 10.3390/children10010002

**Published:** 2022-12-20

**Authors:** Liliane De Sousa Morgado, Kristine De Martelaer, Arja Sääkslahti, Kristy Howells, Lisa M. Barnett, Eva D’Hondt, Aldo M. Costa, Boris Jidovtseff

**Affiliations:** 1Department of Sport and Rehabilitation Sciences, Research Unit for a Life-Course Perspective on Health and Education, CEReKi, University of Liege, 4000 Liège, Belgium; 2Department of Sport Sciences, University of Beira Interior, 6201-001 Covilhã, Portugal; 3Department of Movement and Sport Sciences, Faculty of Physical Education and Physiotherapy, Vrije Universiteit Brussel, 1050 Brussels, Belgium; 4Faculty of Sport and Health Sciences, University of Jyväskylä, 40014 Jyväskylä, Finland; 5Department of Sport, Exercise and Rehabilitation Sciences, School of Psychology and Life Sciences, Canterbury Christ Church University, Canterbury CT1 1QU, UK; 6Faculty of Health, Institute for Physical Activity and Nutrition, School of Health and Social Development, Deakin University, Geelong, VIC 3216, Australia; 7Research Center in Sport Sciences, Health Sciences and Human Development, CIDESD, 5001-801 Vila Real, Portugal

**Keywords:** aquatic skills, water safety, self-perception, tool, motor skills

## Abstract

An international group of experts have developed a pictorial tool to measure perceived water competence for children aged from 5 to 8 years old: the Pictorial Scale of Perceived Water Competence (PSPWC). The aim of the present study was to verify the validity of this tool. In the first part of the study, 120 children were interviewed to investigate face validity of the PSPWC to ensure that all pictorial items were understandable. In the second part of the study, 13 scientific and/or pedagogical international experts were invited to assess the tool’s content validity via an online survey. Face validity results revealed that children were able to understand and sequence correctly the aquatic situations in 92% of the cases. The average Content Validity Index (CVI) of the PSPWC ranged from 0.88 to 0.95, showing acceptable content validity. Feedback from experts and children resulted in a major improvement of the “exit water” situation and minor improvements concerning some other items. Experts confirmed that the PSPWC was globally appropriate for different countries and cultures, except for the situation “water entry by slide” which was not considered usual practice in some countries. The PSPWC opens up to new fields of research; useful both for the prevention of drowning and for the support of children’s aquatic education.

## 1. Introduction

Perceived competence (PC) refers to one’s beliefs about his or her ability to learn and execute specific skills. PC is a key factor in Harter’s competence motivation theory [1]. According to this model, feeling competent is of great importance for motivation and can positively impact cognitive, affective, and behavioral outcomes. One’s PC influences different motivational factors such as the choice to participate in an activity, the attitude and commitment within an activity or even the long-term interest in this particular activity [2,3].

Measuring PC has great scientific and pedagogical interest, especially during early childhood. As Harter [2] stated, children do not perceive themselves as competent in the same way when looking into different developmental areas. Therefore, determining PC in specific contexts and situations seems relevant, especially regarding the self-perception of water competence. Scientists have hypothesized that PC is a key factor in the relationship between fundamental motor skill (FMS) and physical activity (PA) levels during childhood [4,5], and that it would more directly influence one’s intrinsic motivation towards PA than actual skill competence [1]. Therefore, assessing young children’s self-perceived FMS competence has been considered useful to better understand their motivation for PA [4]. Regarding the specificity of self-perception, it is important to develop PC scales that are specific to a motor category and test items, e.g., measuring perception for the same set of skills as an actual motor skill assessment [4,6,7,8].

Water competence (WC) refers to the combination of all personal aquatic movements that help prevent drowning as well as the associated water safety knowledge, attitudes, and behaviors that facilitate safety in, on, and around water [9,10,11]. There is an increasing interest in programs and studies that focus on a broad(er) approach on how children can learn to move and behave safely in the water [10,11,12,13,14].

Water accidents remain a leading cause of unintentional death among children [15,16]. Although the real cause of drowning can vary from one context to another, water competence is a vital aspect for survival [9,13,17,18]. Fear of water is an important factor to consider as associated with avoidance of water, low aquatic skills level and high risk of drowning in children or adolescents [19,20,21,22]. Another critical factor is a child’s self-perception of drowning risk in various aquatic situations. Indeed, it has been suggested that a child’s poor perception of danger [11,12,23] and/or an inadequate perception of their own water competence [12,24,25] can also lead to dangerous behaviors during aquatic activities with increased risks of drowning. Drowning victims’ aquatic skills are generally inadequate or insufficient for survival [11,12,15,26]. Adults’ perception of their children’s water competence is also important, as it can be implicated in inadequate supervision that could lead to dramatic drowning accidents [25,27,28]. Consequently, developing a tool able to assess children’s self-perception of water competence (PWC) could be useful to improve our understanding of drowning accidents and potential misperception of competence levels among children and adults.

Previous research [29] has discussed the difficulty children universally have in understanding their own physical activity and more recently in terms of intensity levels [30], as well as difficulties in estimating their ability to perform physical tasks [31] and specifically difficulties in learning aquatic skills [32]. Additionally, parents have also misperceived their children’s physical activity levels, with most parents of inactive children perceiving that children are sufficiently active and competent, when they are not [33]. On dry land, children’s perception does motivate and/or constrain physical activity behavior so it is reasonable to assume that this would also occur on the water. Recent researches indicate the need for developing such a tool able to assess children’s self-perception to further test this hypothesis in the specific context of water competence. This could be useful for parents, physical education teachers and swimming teachers to better understand how WC is perceived by adults and how closely these perceptions match actual WC; this is vital in reducing the potential mismatch between perceived and actual motor competence, for risky physical activities such as swimming. Such a scale could be administrated and shared in swimming programs with children for teachers to be informed on how children perceive their own aquatic skills, by situating the child in his/her aquatic learning process and identifying acquired and non-acquired skills [34]. So, a PWC tool would both identify the children’s needs and support the teachers in targeting appropriate pedagogical activities in an aquatic environment.

Tools measuring PC are generally based on pictures for children under the age of eight years, because they are more understandable and age appropriate. Pictures also tend to better keep children’s attention and are accompanied by more reliable answers [35]. Previous PC scales constructed for children [4,35] include several drawings for each targeted motor skill, generally ranging from the situation of poorer performance to a better performance. Over the recent years, the use of pictorial scales has grown [4,7,35,36,37]. In fact, these pictorial scales have been used to explore relationships between children’s actual and perceived motor competence and, additionally, to analyze the relationship between children’s perception of their motor competence and their levels of physical activity [36]. Recently, an Australian group developed a self-report scale to assess children’s self-perceived physical literacy [38]. However, to our knowledge only one study has developed a specific scale for the perception of competence linked to aquatic activities in children: the Perceived Aquatic Competence Pictorial Scale (PACPS) [7,22]. This tool consisted of 18 items focusing on aquatic motor skills and children’s attitude towards water linked to recreation and activities. Although the psychometric qualities appear satisfactory, this PACPS tool does not fully assess all aquatic fundamentals and is only aimed at children aged 4 to 5 years old. Therefore, there was a need to develop a tool for assessing children’s perceived water competence in a more comprehensive way, making it appropriate for a wider age spectrum.

It was based on this need that, in 2016, scientific experts from the Early Years Special Interest Group (SIG) of AIESEP (International Association of Physical Education in Higher Education) decided to work on the development of a Pictorial Scale of Perceived Water Competence (PSPWC). The PSPWC aims to address the following specific needs: (i) to be accessible for children aged from 5 to 8; (ii) to be suitable for different swimming levels; and (iii) to cover all the aquatic fundamentals. The building and the description of the PSPWC has been detailed in the testing manual [39]. The purpose of the present study is to examine the face validity of the PSPWC tool when used with Belgian children as well as the content validity of this pictorial scale.

## 2. Materials and Methods

### 2.1. Instrument Development

The PSPWC tool is the product of a collective reflection, which started within the “AIESEP Early Years SIG” in 2016. Within this international group, there is complementary expertise in the field of water competence, motor competence, and the assessment of perceived competence. Initial discussion aimed to determine the design of the pictorial scale in order to meet three specific aims, as mentioned above in (i), (ii) and (iii). Different aquatic situations/skills with gradual difficulties were selected in order to explore fundamentals that are required for becoming water safe, learning to swim and drowning prevention [12,15,40,41]. These aquatic fundamentals were initially: water entry, water exit, immersion, water orientation (or balance), buoyancy, propulsion and breath control competencies. Two additional aquatic fundamentals were added later on: gliding and vision (open eyes under water). Because of the target age (i.e., 5 to 8), all aquatic skills were represented by drawings to: (1) induce young children’s interest, (2) maintain their attention, (3) facilitate their understanding independently of language skill, and (4) get meaningful responses [2,4]. A professional illustrator was selected by the expert group, to achieve the images/pictures based on detailed movement descriptions.

The pictorial scale was constructed based on a three-level progression for each aquatic situation/skill, as follows: Level 1 = “not able to perform the skill”; Level 2 = “partly able to perform the skill, and thus in progress”; Level 3 = “able to perform the skill”. This was preferred to a dichotomous “able vs. not able” approach as considered much more appropriate with a process-oriented scale representing a child’s developmental progression [40]. Such a format may also minimize the likelihood of children giving a socially desirable response [2,7] and has been previously used in another aquatic pictorial scale [7].

The very first version of the PSPWC consisted of 16 different situations corresponding to 16 aquatic skills and was submitted to all members of the expert group for critical analysis, who assessed the relevance of each aquatic situation and the quality of the pictures representing the three levels per skill. Only small changes were made to the pictures to make them more realistic and more understandable. Examples of these changes were the depth of the water in the exit situation, and the removal of supporting adults in the intermediate level, being replaced by a floating device. A seventeenth situation was added to investigate perceived competence in transverse axis rotation.

### 2.2. Instrument Translation (from English to French Language)

To maintain the content validity of the PSPWC in different countries, the instrument needed to be well translated linguistically, and adapted culturally, if needed [42]. While international content validity has been achieved with the English version of the PSPWC, it was essential to have a French translation of the instrument for the face validity. A French translation of the PSPWC was made according to recommendations [42]. The instrument was translated from English to French language by two translators, then a back translation to English was completed by two English native speaker specialists, who did not have access to the original English version. Discrepancies were resolved in accordance with translators, to result in the final French version that was used for both Study 1 and Study 2 (see below). This procedure has been detailed in the Manual of the PSPWC [39] for replication of the procedure for future international teams. Identical translation procedures can be used in view of future cross-country and/or cross-cultural comparisons.

### 2.3. Preliminary Face Validity (Study 1)

A preliminary face validity check was conducted with children as recommended [4,43] to check their understanding of the pictures. This preceding/first step involved 50 Belgian children (50% girls) aged 4 to 8 years old (M = 6.1, SD = 1.4). The interviews were conducted in two French speaking schools in Belgium and under the same conditions as described in Study 2 (see below). Children identified elements that led to the misunderstanding of the pictures and, therefore, confusion in the ordering from least competent to most competent. Seven aquatic situations/skills had to be improved and four illustrations needed to be modified. The PSPWC was adapted according to these results in order to improve children’s understanding [39]. This preliminary investigation also showed that Belgian children aged under 5 years had less understanding of the scale and this provided a rationale to focus on children aged from 5 to 8 years’ old. At each stage of the building/development process, the tool underwent several adaptations considering what had previously been observed on the field and/or discussed within the group of experts. This process led to the revised version of the PSPWC that was used in Study 2 and that is described in Table 1.

### 2.4. Actual Face Validity (Study 2)

#### 2.4.1. Participants

One hundred and twenty Belgian children (55.8% girls), aged from 5 to 8 years (M = 6.5, SD = 1.1), took part in the face validity examination. All age groups were of equal size: 5-year-olds (n = 30, 46.7% girls); 6-year-olds (n = 30, 60% girls); 7-year-olds (n = 30, 53.3% girls) and 8-year-olds (n = 30, 63.3% girls). All children were from three voluntary participating schools of the French speaking part of Belgium. Ethical approval was granted by the Ethical Committee of the University of Liège, and permission was obtained from the respective school authority; parents or guardians provided written consent and children assented. To participate in the study, children had to speak French fluently to ensure a good understanding of the questions addressed during the face validity procedure. Swimming experience and level were not considered as inclusion criteria.

#### 2.4.2. Face Validity Procedure

Individual structured interviews with the children were used to assess face validity of the PSPWC [4,8]. These interviews were organized in an empty quiet place in the children’s respective school during class hours. Each interview was audiotaped and always conducted by the same researcher (LM). The interview started with a short and standardized explanation of what would take place. This explanation was presented to the children as a “pirate story” in order to get their attention (see Figure A1 in Appendix A). Once the child understood what to do, and upon his/her agreement to participate, the evaluator started the questionnaire.

The questionnaire was based on the method described in Barnett et al. [4], and included three standardized questions adapted to each aquatic situation/skill: (1) The knowledge of the situation: Q1 = “Do you know what is this aquatic situation?”; (2) the ability to sequence the pictures correctly: Q2 = “Can you place these pictures in order from the one showing a child not able to perform the skill, to the one showing a child able to perform it”?; and (3) the reasoning of the sequence: Q3 = “Can you explain why you put the pictures in that order?”. If the child was not familiar with the aquatic situation, the evaluator showed a short video illustrating the skill performed in its more advanced form. A binary scoring system was used to measure children’s familiarity with the aquatic situation (i.e., 0/1 = children do not know/know the situation). Situation familiarity corresponds to the proportion (in %) of children who knew the situation. To investigate the ability of the children to find the right sequence, the evaluator gave them the three pictures representing the different levels of a given aquatic situation/skill in a randomized order and asked the child to classify the pictures from “not able” to “able”. A binary approach (i.e., 0/1 = not correct/correct) was also used to score the ability of the children to sequence the pictures in the desired order, allowing the determination of the right sequencing proportion (in %) among all children for a given skill. Regarding the last question, the evaluator verified if the child understood each of the three pictures by letting them provide a logical reasoning for their sequencing. One point was attributed for each picture per skill level so that the global understanding for each aquatic situation was rated from 0 (= any picture understood by the child) to 3 points (= all three pictures understood by the child). Sequence explanation score (in %) was obtained by dividing the average score of all children by three. In addition to this, the evaluator noted if a child’s arguments were based on a good understanding of the skill progression between the pictures and/or other factors such as emotional expression of the child illustrated in the pictures itself. An encoding grid (see Table A1 in Appendix B) was used during the interview, to collect data in a repeatable manner with every child.

### 2.5. International Content Validity

Consulting a panel of experts is a common method used to determine the content validity of a newly developed tool [44,45]. It provides an external point of view about the clarity of each of the test items as well as information on the relevance of the tool and specific suggestions for improvements [46]. A minimum of five experts is recommended to have sufficient control over chance agreement [47]. However, other authors have suggested to include several experts, ranging between 6 to 20 [48]. In the present study, the consulted expert panel included 13 international experts (i.e., two from Australia, four from Belgium, three from Finland, two from Portugal, and two from the United Kingdom). The inclusion criteria for experts were: (1) to have published as a scientist or/and to have worked in the field as a teacher in children’s water competence or/and motor skill assessment, (2) to have at least five years of experience in this area, and (3) not being involved in the PSPWC instrument development.

These international experts were chosen by convenience and individually contacted by email, to gain their participation approval and to explain to them: (i) the objective of the study; (ii) how the instrument was developed alongside a description of the instrument; (iii) the intended target population; and (iv) the reason why they have been chosen as an expert to contribute. Experts who accepted to participate were asked to complete an online survey presenting the PSPWC in detail and questioning critical aspects on each test item as well as on the total scale. The survey was developed following the recommendations of Alexandre and Coluci [49]. It included four closed questions per aquatic situation/skill, all based on 4-point Likert scales: (1) relevance of the situation: Q1 = “How relevant do you rate this situation?” with a Likert scale from very irrelevant to very relevant; (2) clarity of each level description: Q2 = “How clear is the detailed description of each level?” with a Likert scale from very unclear to very clear; (3) agreement with progression: Q3 = “Do you agree with the presented progression?” with a Likert scale from strongly disagree to strongly agree; and (4) understanding/representativeness of each picture: Q4 = “How understandable (representative) are the pictures?” with a Likert scale from very poor to very good. The survey also included four closed questions concerning the overall scale: Q5 = “This pictorial scale is adapted to the children from 5 to 8 years old”; Q6 = “This pictorial scale is well illustrated”; Q7 = “This pictorial scale is able to assess children water competence level”; and Q8 = “This pictorial scale is relevant and complete” with a Likert scale from strongly disagree to strongly agree.

At the end of the online survey, experts were allowed to give general feedback on the PSPWC (Q9 = “Feel free to add any general comments or suggestions that could help us to improve the pictorial scale”). Results of the survey were exported to an Excel spreadsheet for data analysis.

### 2.6. Data Analysis

All statistical analysis was achieved with both Excel (Microsoft, Washington, DC, USA) and Statistics software (Version 13.2, TIBCO Software Inc., Arlington, VA, USA). Level of situation familiarity, sequencing and sequence explanation were calculated for each aquatic situation/skill and on average. Non-parametric Spearman’s rank correlation was used to assess relationships between averaged variables and a Kruskal–Wallis test was used to investigate age effect on each variable. A Mann–Whitney U test was also used to investigate between age group differences. Statistical significance was set at *p* < 0.05

As criteria for assessing face validity were not clearly defined in previous studies [4,8], it was decided to use the following interpretation rules: percentages between 0% and 49% are considered as very low, between 50% and 69% as low, between 70% and 89% as moderate, from 90% as high and 100% as perfect.

Content validity of the PSPWC by experts was assessed through the Content Validity Index (CVI), which was calculated for each question per aquatic situation/skill. Individual CVI corresponded to the proportion of experts giving item a relevance rating of 3 or 4 in Likert scales [50]. To get an overview of the validity of the whole pictorial scale, we averaged the 17 CVIs of each situation. According to the high number of experts involved (i.e., 13), the acceptable cut-off score of CVI was set at 0.77, implying that 10 out of the 13 experts agreed on the item. Such a cut-off score is in line with what was suggested in previous studies [43,50,51].

## 3. Results

### 3.1. Face Validity (Study 2)

The results of the face validity are presented in Table 2 and revealed that on average, 74% of the children were familiar with the different situations. Six out of the seventeen aquatic situations/skills (i.e., skills 1, 2, 14, 15, 16, 17) presented a very low (<50%) to low (50–69%) level of familiarization. Lying down in a prone position (skill 1); vertically treading water (skill 14) and longitudinal axis rotation (skill 15) were known by less than 50% of the children (see Table 2).

Children’s ability to sequence the three pictures correctly was high on average (92%). Situation/Skill 13 (i.e., water exit by climbing out) was correctly sequenced only by a low proportion (68%) of children. The ability of the children to explain the sequence of a given situation with logical reasoning was high, on average (94%). Similarly, water exit by climbing out was the more difficult aquatic situation for children to provide reasoning and explanation for sequencing the skill levels (78%).

When children were asked to explain sequencing, two types of argument were noticed. Mostly, explanations were based on the differences in aquatic skills they perceived from one picture to another. In a few cases, their argument was based on the facial expression of the child illustrated in the pictures. The illustrated child with a floating device and/or a negative emotion (fear) were successfully identified as a low level of performance, whereas a positive emotion (joy) was identified as a high level of performance.

Age effect on face validity was investigated through Kruskal–Wallis and Mann–Whitney U tests as reported in Table 3. Five-year-old children reported a significantly lower ability to sequence the different levels per aquatic situation/skill correctly in comparison with children aged from 6 to 8 years (*p* < 0.001). A statistically significant difference (*p* < 0.001) was also observed between 5-year-old and 8-year-old children for the sequence explanations (i.e., able to explain logically the sequence and to describe the 3 different skill levels of the same aquatic situation).

Relationships between averaged variables are presented in Table 4. A moderate positive correlation was observed between children’s sequence explanation and their ability to sequence correctly (r = 0.62, *p* < 0.001). A low positive correlation was observed between age and the ability to sequence correctly the three pictures (r = 0.43, *p* < 0.001), age and sequencing explanation (r = 0.37, *p* < 0.001), and between situation familiarity and the ability to sequence (r = 0.44, *p* < 0.001).

### 3.2. Content Validity

Results of the content validity are presented in Table 5. All Content Validity Indexes (CVIs) related to the relevance of the situation reached the predetermined acceptable cut-off score of 0.77 with an average of 0.95. Lower relevance of the description was reported by the experts for skill 7 (i.e., water entry by slide) and skill 16 (i.e., sagittal rotation). The relevance of the detailed description of the skills and their different levels showed an average CVI of 0.92, with individual values ranging from 0.85 to 1.00. Agreement with the progression of the different skill levels revealed an average CVI of 0.90, with high agreement for each of the skills (i.e., >0.80), except for skill 16 (i.e., sagittal rotation, CVI = 0.77). For the representativeness of each picture/skill level, CVIs were on average lower in pictures displaying levels 1 (0.88) and 2 (0.87) as compared to the picture displaying level 3 (0.95). Lower CVIs (≤0.77) were reported in some pictures of 6 aquatic situations (i.e., back star, front star, exiting deep water, longitudinal rotation, sagittal rotation, and transverse rotation). The average CVI of the PSPWC showed that its content validity can be considered as acceptable with values ranging from 0.88 to 0.95.

The CVI was also determined for the questions concerning the total pictorial scale. Experts agreed that the scale was suitable for 5 to 8 years old children (CVI = 0.92); well-illustrated (CVI = 1.00) and able to assess perceived water competence (CVI = 0.92). A lower but acceptable CVI (0.77) was reported when experts were asked if they consider the PSPWC as relevant and complete. At the end of the questionnaire, the experts had the opportunity to leave a comment to justify their answer. Three of them considered that the inclusion of emotions on the drawings could mislead children. Two experts criticized the use of floating devices in the intermediate level. Finally, one expert wished that the tool was more associated with the positive pedagogy of success and criticized the fact that Level 1 was associated with a child’s failure.

## 4. Discussion

The present study investigated the face and content validity of the PSPWC in children aged from 5 to 8 years old. Examining face validity is a very important process in the validation of instruments of this type because understanding how the pictures are viewed and understood by children can show the elements that lead to the misunderstanding of the different situations or the difficulty in sequencing skill levels [4,52]. Globally, our face validity analysis confirmed that the PSPWC was an appropriate tool to assess self-perceived water competence in children from 5 to 8 years old. We showed that the comprehension of the pictures as well as the sequencing were satisfactory in children of 5 years and were excellent from 6 years of age onwards. Content validity from an international panel of experts confirmed that the PSPWC was globally appropriate to different countries and cultures, except for the aquatic situation/skill “water entry by slide” (skill 7) that was not considered as usual practice in some of the countries involved, and therefore can be considered as optional in the scale.

Four-year-old children were not included in the present study because an unsatisfactory level of understanding of the pictures was found in the preliminary study in this particular age group [39]. This does not mean that we cannot use this tool with this younger population, but that we should be aware that the results can be biased by the lower understanding of the pictures. In contrast, De Pasquale et al. [34] reported that the PSPWC was appropriate for Australian children aged 4–8 years who had sufficient understanding of the outlined aquatic skills. One reason, which may explain the difference in the understanding of pictures between Australian and Belgian four-year-old children, could be swimming experience and the familiarization of the children with water and with the aquatic situations presented in the PSPWC. Australian children were recruited within swim centers, while children in our research were recruited in schools thereby including children who are and who are not familiar with aquatic recreation. This difference may explain why face validity for 4-year-olds was considered as acceptable in the Australian study while it was not the case in our preliminary face validity (Study 1). This is in line with the moderate significant correlation observed in the current study between situation familiarity and ability to explain (Table 4).

The children in the present study sample reported a moderate familiarization with different situations (74%), a high ability to sequence (92%) and a high ability to explain correctly (94%) the aquatic situations and their pictures/skill levels. This means that even if the aquatic situations/skills were not familiar to the children, their illustration made them very understandable. A more detailed approach revealed that some of the situations were either less known or less understood by the children. Children who did not report a high level of familiarity with skill 1 (i.e., lying down in a prone position) and skill 2 (i.e., standing in the water) justified themselves by saying they had swimming lessons in pools where the water depth did not allow the practice of these skills and hence were not familiar with them. Skill 14 (i.e., treading water) and the skills that involved changing direction (i.e., longitudinal rotation, sagittal rotation, and transverse rotation) also revealed low percentages that can be justified as these skills are identified in the literature as more advanced skills [12,14,40] and so the development or practice of these skills in this targeted younger age group is less common.

When the children were asked to sequence the pictures, some revealed difficulty in three skills, namely “lying down in a prone position” (skill 1), “blowing bubbles” (skill 3) and “exiting deep water” (skill 13). It is important to note that certain children were able to sequence the pictures correctly but not able to explain why, while others were able to explain the logic of the sequence performed and to describe the pictures correctly but could not sequence them correctly. For example, in “lying down in a prone position” situation (skill 1), some children considered that being on all fours (i.e., picture/level 2) was easier than standing in the shallow water (i.e., picture/level 1) and failed to present the expected sequence. In the “blowing bubbles” situation (skill 3) a few children reported difficulty in understanding picture/level 1, they identified the ping-pong ball as being a ‘bubble’ in pictures/levels 2 and 3. This lack of understanding led some children to reverse the order of picture/level 1 and 2. Exiting a deep water situation (skill 13) proved to be unclear for children as it was very difficult for them to explain the logical reasoning of the sequence. Some of the children (n = 12) switched between picture/level 2 and 3 as the child illustrated in the most advanced level seems to be in a greater difficulty than in level 2. In the other situations, the child drawn in level 3 never appears to be in difficulty. This difference diverted the attention of the children who classified the images according to the apparent difficulty of the child rather than the actual difficulty of the task. Previous study by De Pasquale et al. [34] revealed that some children had difficulties in sequencing some situations (i.e., “lying down in a prone position” [skill 1], blowing bubbles [skill 3] and exiting deep water [skill 13]), meaning that some illustrations may be unclear and need some improvements.

Our face validity also revealed some confusion in sequencing the “back star” situation (skill 5). The reason was that the child’s legs in picture/level 3 showed instability in comparison with picture/level 2 and, as a consequence, few children tended to classify this third level picture as the intermediate level despite the absence of a floating device. By contrast the absence of the floating device in the picture/level 1 of “water entry by slide” (skill 7) also brings confusion with the picture/level 2 for a limited number of children (8 out of 120). While these issues had no significant consequence on the sequencing score, it is important to pay attention to these details in order to improve the quality of the tool. In fact, according to Harter and Pike [35], the graphical representation facilitates the understanding of the skill to be performed. If the illustration confuses or does not highlight the main elements allowing the skill level to be judged [4], it is possible for the child to focus on secondary and non-descriptive elements that can impact their understanding of the sequences. This was the case with the “water exit climbing” situation. These results of the face validity led to pictorial improvements with the aim to reduce such confusion for the children (see Table A2 in Appendix C).

The quality of the illustration is not the only factor that may influence a child’s understanding of pictures and the correct understanding of the sequences. One’s chronological age is also a determining factor in face validity. In fact, previous studies revealed that children of preschool age had more difficulty expressing their understanding of pictures [4,34,39]. The same was found in the present study, where children who were 8 years of age could explain the pictures more easily than children of 5 years of age. Additionally, there was a significant difference in the sequencing of pictures between children aged 5 years and children aged 6 to 8 years. Spearman’s rank correlation analyses showed that this ability to sequence pictures in the right order according to skill progress is more correlated with the ability to explain, rather than with age. This interesting result suggests, in accordance with the current knowledge on the cognitive development of the child [53], that cognitive maturation is more important than chronological age for being able to create a logical sequence of pictures.

According to Lopes et al. [8], familiarity with a given skill may influence the ability of the children to understand and sequence its illustration. Effectively, our results revealed that there is a significant moderate and positive correlation (r = 0.44) between the knowledge of a given skill and the ability to describe/perceive the pictures. It can be hypothesized that the validity of the PSPWC would be higher in a population that has already been familiarized with the aquatic environment and, more specifically, with the presented aquatic situations/skills. All children implicated in the present face-validity were coming from a high income and high educated country equipped with numerous swimming pools. In Belgium, learning to swim and water competence development is included in the school curriculum. Further research is needed with low- and middle-income countries in which access to education and a swimming pool are limited to privileged populations. Research is needed to understand if perceived water competence can be measured in a reliable way with all children, regardless of their level of aquatic experience and competence. While internal consistency of the PSPWC has been recently validated in a very similar population [25], its reproducibility has to be investigated between age groups and aquatic/swimming experience. Further results should investigate variable conditions in which the PSPWC could be used in a reliable way with children.

Examining the content validity was another important step in the PSPWC validation process aimed to obtain pertinent comments and opinions from international experts. Having a panel of experts coming from different countries was important, as one of the aims of the PSPWC was to allow cross-country and/or cross-cultural comparisons. It should be noted that these experts all came from high-income countries where the conditions for water education are favorable. These results cannot be extended to low-income countries without prior scientific verification. The content validity of the instrument was found to be globally good, with most of the reported CVIs above the predetermined and recommended value of 0.77 [43,51]. A high rate of experts agreed that the instrument was appropriate for children from 5 to 8 years old, well-illustrated and able to assess perceived water competence. These results were in accordance with the global positive feedback reported by swim instructors about the PSPWC in Pasquale et al.’s [34] study. Among the 17 aquatic situations/skills, 14 items were considered very relevant (CVI > 0.90). Only skill 7 (i.e., water entry by slide) and skill 16 (i.e., transverse rotation) revealed a lower but acceptable CVI of 0.77.

While most of the experts considered the “water entry by slide” (skill 7) as very relevant, three of them rated it as not relevant. This aquatic situation was suggested to be included in the PSPWC by Belgian experts, as in that country it is frequently used to improve children’s water entry. Some experts from other countries mentioned that it was less common in their countries but accepted its integration in the scale. Our results confirmed that this situation is not a common aquatic situation/skill in all countries and revealed international cross-cultural differences. In the Australian research [34], some swim teachers even suggested to remove this situation. With regards to these results, the situation “water entry by slide” would have limited relevance in any cross-cultural comparisons and could be considered as optional. In addition, the PSPWC has two other situations assessing water entry (i.e., skill 11 “jump into the water” and skill 12 “dive into the water”) and would probably not be impacted by removing aquatic situation/skill 7.

Sagittal rotation (skill 16) was not considered relevant by three of the 13 experts. According to these experts, this rotation was not naturally used by the children: “My opinion is that this skill is not relevant. Little swimmers turn around horizontally, not vertically”. This sagittal rotation was included in the scale as water orientation is a very important aquatic fundamental especially in drowning prevention [12] and when developing the tool, we wanted to investigate it around all three body axes. In addition, the face validity showed that children were able to sequence and explain all the rotation skills on a similar way. However, further research is needed to investigate if these situations are redundant and can be reduced to most relevant one to be determined.

A notable proportion of the experts (77%) considered the PSPWC as relevant and complete. Such a result is not surprising, since the higher the number of specialists, the lower the agreement between them. It is more difficult to obtain full agreement about an item with thirteen experts than with two [46]. The three experts who considered that the PSPWC was either not relevant or not complete enough, justified their scores with four main points. Two of them suggested a scale should be developed with more sub-levels for each situation. From a pedagogical point of view, it is always interesting to have a more detailed sequence of progression; however, the more sub-levels there are, the more difficult illustration becomes, and also the more difficult for a child to complete. The “3-step progression” approach selected in the PSPWC was considered by the instrument developers to be a good compromise as it allowed the tool to be easily understood by the target age group and to be process-oriented [40]. Such a three-level approach was also used in a previous aquatic pictorial scale [7].

Three experts were concerned by the integration of emotional components in the description of the aquatic situation and on the pictures. According to these experts, such an approach should be questioned because the emotion expressed by a child is not necessarily related to the level of aquatic competence. An expert underlined that from a pedagogical point of view it is important to present each level in a positive way. In accordance with these comments, emotional facial aspects were deleted from any written description of the situations. However, our face validity (Study 2) confirmed that emotion, and especially the child’s apprehension illustrated in level 1, was useful for children’s understanding and it was decided not to remove this component but to reduce it on the pictures, as presented in Table A2 in Appendix C.

Two experts were concerned by the use of flotation devices in several situations as they consider these aids should not be considered as the only effective method to learn to swim. While one can agree with this pedagogical point of view, the use of a floating device was considered as an understandable and reliable way to represent a child in progression. Using a similar approach in different situations was considered an important component for global understanding and unity of the PSPWC. It was also reported that the presence of a face expression and a floating device helped the children to correctly identify why certain pictures represented a child better performing a skill and recognized which pictures represented the child in each stage/level of skill execution [34].

The combination of both face and content validity allowed us to identify the pictures of the aquatic situations/skills that generated difficulty in comprehension and/or difficulties in the sequencing the progression. Improvements implemented to the picture are reported in Table A2 in Appendix C. Minor changes to the tool presentation were also achieved in order to erase any discordances between the description of the aquatic situation/skill and the corresponding pictures according to the three levels of progress. The last version of the PSPWC, including the above-mentioned improvements is presented in a test manual and freely available online [39].

Some limitations to the PSPWC presented in this validation study need to be highlighted. The tool focuses more on fundamentals of water competence and less on advanced aquatic skills such as swimming style, meaning that a ceiling effect, already observed in previous studies [25,34], appears in the most advanced children. Pictures are representing a very specific context (i.e., a slim Caucasian boy in a swimming pool) and could be diversified to be more adapted to both genders, to all ethnic populations, morphologies, and environmental conditions. The tool was tested without gender balance, in high income settings, with Caucasian pictures and validated by experts from high-income countries. The usefulness of this tool outside of this setting/context has to be investigated. In addition, the tool focused mainly on the specific competences developed in the swimming pool and not on the larger concept of water competence as defined by Langendorfer and Bruya [40] in relation to drowning prevention, which also includes knowledge of local hazards and attitude and values toward water environment [12,14]. Further development, including contextualized pictures and additional water safety situations, is thus needed to investigate PC in open water settings. The population included in the study could be considered a limitation as they were not from randomly selected but voluntary schools and could not be considered as representative of the general (pediatric) population. Our study was limited to children from 5 to 8 years old while another research also included younger children [34]. Yet, our choice of ages was influenced by our preliminary research showing lower face validity in the youngest population.

Another potential limitation is the time taken for assessment of all situations. An adaptation of this instrument (the PSPWC-short form) has since been developed and tested in over 100 Australian children using only four of the seventeen situations [54]. The results of this study, currently being published, are encouraging: moderate positive correlations were found between actual swim level and perception of: retrieving an object in deep water (rho = 0.57), swimming on front (rho = 0.60), swimming on back (rho = 0.69), and treading water (rho = 0.63), and also, the summed score of all four (rho = 0.71). Further studies should verify correlations with full PSPWC scores.

The strengths of this study include the complementary approach of examining both the PSPWC’s face and content validity with critical information issues from children themselves as well as a panel of international experts. Furthermore, our sample size of children was superior to similar studies, increasing the power of our results. The fact that children were recruited in public schools permitted the inclusion of children who were not enrolled in any swimming schools, which was not the case in the two first published studies using the PSPWC [25,34]

It is important to investigate the relevance of the tool in the most representative population and not only in a population familiarized with water competence, as previous experience with the aquatic environment might influence children’s self-perception. Another strength of the research was the use of an international panel of experts coming from five different countries (i.e., Australia, Belgium, Finland, Portugal and UK). This approach was important to ensure that the tool could be validated for different countries, creating a window of opportunity for international cross-cultural research.

Determining the validity and reliability of an instrument is a complex process that requires several steps. This face and content validity research was an important step in this process and further research should investigate its construct validity and test–retest reproducibility.

Further research should verify if the tool can offer reliable information and additional studies are needed in large and more variable populations to confirm previous results on the relationship between perceived and actual competence [25]. The development of the PSPWC opens up new avenues for national and international research on water competence useful both for the prevention of drowning and for the support of children’s aquatic education.

## 5. Conclusions

The results of this validation study confirmed that the initial version of PSPWC used in the present study had acceptable face and content validity and was appropriate to access perceived water competence in children aged 5 to 8 years. Feedback from children and experts resulted in a major improvement of the “exiting deep water” situation (skill 13) and in minor improvements concerning the presentation of the aquatic situations and the skill level pictures. Content validity from an international panel of experts confirmed that the PSPWC was globally appropriate to different countries and cultures, except for the aquatic situation/skill “water entry by slide” (skill 7) that was not considered as usual practice in some of the countries involved, and therefore can be considered as optional in the scale. Further researches are needed to validate the PSPWC in low- and middle-income countries and in diverse learning settings/contexts.

## Figures and Tables

**Table 1 children-10-00002-t001:** Aquatic fundamentals measured by the PSPWC.

		Aquatic Fundamentals								
Aquatic Skills	Depth of Water	Water Entry	Water Exit	Immersion	Water Orientation	Buoyancy	Gliding	Propulsion	Breath Control	Vision/Looking under Water
Sk1—Lying down in a prone position	SW			X	X			X		
Sk2—Standing in the water	SW to WSL			X						
Sk3—Blowing bubbles	WHL			X					X	
Sk4—Catching an object	WHL			X	X				X	X
Sk5—Back star	WHL or WSL				X	X				
Sk6—Front star	WHL or WSL				X	X			X	
Sk7—Water entry by slide	WSL	X								
Sk8—Gliding under water	WHL or WSL				X	X	X		X	
Sk9—Leg propulsion on the back	WSL-DW					X	X	X		
Sk10—Leg propulsion on the front	WSL-DW					X	X	X	X	
Sk11—Jump into the water	DW	X								
Sk12—Dive into the water	DW	X					X		X	
Sk13—Exiting deep water	DW		X							
Sk14—Treading water	DW				X	X				
Sk15—Longitudinal rotation	DW				X	X		X		
Sk16—Sagittal rotation	DW				X	X		X		
Sk17—Transverse rotation	DW				X	X		X		

Different depths of water: shallow water (SW) (i.e., water at knee level); water at hip level (WHL), water at shoulder level (WSL); deep water (DW) (i.e., head fully under water in standing position).

**Table 2 children-10-00002-t002:** Face validity of the PSPWC: situation familiarity, ability to sequence and to explain the sequence by the children.

Aquatic Skills	Situation Familiarity ^1^	Sequencing ^2^	Sequence Explanations ^3^
Sk1—Lying down in a prone position	14%	87%	90%
Sk2—Standing in the water	68%	93%	93%
Sk3—Blowing bubbles	90%	89%	93%
Sk4—Catching an object	93%	95%	95%
Sk5—Back star	82%	93%	97%
Sk6—Front star	91%	95%	96%
Sk7—Water entry by slide	96%	93%	97%
Sk8—Gliding under water	71%	95%	97%
Sk9—Leg propulsion on the back	92%	98%	98%
Sk10—Leg propulsion on the front	93%	98%	98%
Sk11—Jump into the water	96%	97%	98%
Sk12—Dive into the water	72%	95%	95%
Sk13—Exiting deep water	94%	68%	78%
Sk14—Treading water	38%	93%	96%
Sk15—Longitudinal rotation	43%	92%	92%
Sk16—Sagittal rotation	66%	93%	89%
Sk17—Transverse rotation	58%	97%	92%
Average	74%	92%	94%

^1^ Situation familiarity = % of children who know the aquatic situation. ^2^ Sequencing = % of correctly positioned pictures according to skill level. ^3^ Sequence explanations = ability to explain logically the sequence and to describe the 3 different level pictures of the same aquatic situation. Percentage interpretation: percentages between 0% and 49% are very low, between 50% and 69% are low, between 70% and 89% are moderate, from 90% are high and 100% represents perfect.

**Table 3 children-10-00002-t003:** Situation familiarity, sequencing and ability to explain the sequence for each age categories.

	5 Years Old	6 Years Old	7 Years Old	8 Years Old	Age Effect (*p*-Value)
Situation familiarity	67%	76%	75%	77%	NS
Sequencing	81% ^6, 7, 8^	96%	95%	97%	*p* < 0.001
Sequence explanations	87% ^8^	97%	97%	99%	*p* < 0.001

Significant difference between and age groups are presented in the table with ^6^ (difference with 6 years old); ^7^ (difference with 7 years old) and ^8^ (difference with 8 years old).

**Table 4 children-10-00002-t004:** Spearman rank correlations between variables.

	Age	Situation Familiarity	Sequencing	Sequencing Explanation
Age	1.00	0.19	0.43 *	0.37 *
Situation familiarity		1.00	0.29	0.44 *
Sequencing			1.00	0.62 *
Sequencing explanation				1.00

* Significant correlation at *p* < 0.01 level.

**Table 5 children-10-00002-t005:** Individual CVIs by item and question and average CVIs by question.

	Content Validity Index (CVI) Results					
Aquatic Skills	Relevant Situation	Relevant Description	Agreement with the Progression	Representativeness of Each Picture (n = 13)		
				Picture 1 *	Picture 2 *	Picture 3 *
Sk1—Lying down in a prone position	1.00	0.92	0.92	0.92	0.92	1.00
Sk2—Standing in the water	1.00	0.92	0.92	0.85	0.85	0.92
Sk3—Blowing bubbles	0.92	0.92	0.85	1.00	1.00	0.85
Sk4—Catching an object	1.00	0.92	0.92	0.92	0.85	0.92
Sk5—Back star	1.00	0.92	0.92	0.69	0.77	1.00
Sk6—Front star	1.00	1.00	0.92	0.77	0.85	1.00
Sk7—Water entry by slide	0.77	0.85	0.85	0.85	0.85	0.85
Sk8—Gliding under water	0.92	0.92	0.92	0.92	1.00	1.00
Sk9—Leg propulsion on the back	1.00	0.92	0.92	0.92	0.92	1.00
Sk10—Leg propulsion on the front	1.00	1.00	0.92	0.92	0.92	1.00
Sk11—Jump into the water	1.00	0.85	0.92	0.85	0.92	1.00
Sk12—Dive into the water	1.00	1.00	0.92	0.92	0.92	1.00
Sk13—Exiting deep water	0.92	0.92	1.00	1.00	0.77	0.77
Sk14—Treading water	1.00	0.92	0.92	0.85	0.85	1.00
Sk15—Longitudinal rotation	0.85	0.85	0.85	1.00	0.77	0.85
Sk16—Sagittal rotation	0.77	0.92	0.77	0.77	0.85	0.92
Sk17—Transverse rotation	0.92	0.92	0.85	0.85	0.77	0.92
Average CVI of the PSPWC	0.95	0.92	0.90	0.88	0.87	0.95

* Picture 1~Level 1 = “not able to perform the skill”; Picture 2~Level 2 = “partly able to perform the skill, and thus in progress”; Picture 3~Level 3 = “able to perform the skill”.

## Data Availability

Not applicable.

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
