# Peer review of "Face and Content Validity of the Pictorial Scale of Perceived Water Competence in Young Children"

_children, 2022, doi:10.3390/children10010002_

Round 1

Reviewer 1 Report

Thank you for this paper.  

The abstract is well done.

Introduction:

consider new paragraph at line 58 starting with Water accidents---

consider adding the concept of injury risk self perception in lines 67/68.

lines 69-75 - there is debate/uncertainty about the use of a visual thought activity to inform risk related physical activity choices.

Materials and Methods

consider more paragraphing in line 106-138 to improve ease of reading

consider removing the word "First" in line 157.

Results

Discussion

Line 448 - 451 consider adding that further research is needed with low-and-middle income countries or state that all of the children reviewing this tool have been in high income nations.  

Line 452-456 states the international experts determine that the tool is "globally good" when all of the experts may be from high income states and possibly from high educational settings.  Therefore consider stating that the tool was considered helpful/useful by the international experts, who represent high income nations and high levels of education. 

Lines 527-531 need to be considered for further magnification and or more clearly stating that this tool was tested without gender balance, in high income settings, with caucasian graphics and primarily caucasian review group.  The usefulness of this tool outside of this setting/context is not known.

Lines 560-563 - all the experts are high income nation context.

Conclusion

Line 575 - the validity of this tool is not determined outside of high income countries so a more correct statement would be considered here as - needs further validation in LMIC's and in diverse international settings/context.

Author Response

Reply in the document

Reviewer 2 Report

The purpose of this study was to investigate the face validity of the PSPWC tool as well as the content validity of this pictorial scale. The strength of this work is that it provides interesting findings in the field of swimming didactics. Namely, the results confirm that the PSPWC has acceptable face and content validity and is suitable to capture the perceived water competence of children aged 5 to 8 years. The research problem is clearly presented and the authors used appropriate methods to address it. The results are presented accurately. The paper ends with a detailed discussion in which previous studies have been considered. The authors have done an excellent work. I congratulate them. However, there are some specific suggestions to improve the manuscript:

·       To remove the abbreviation PSPWC from the title.

·       To use the keywords you did not use in the title .

·      In the context of perceived ability (competence, knowledge), fear of water is another important factor for novice swimmers that can be perceived. Recently, the instrument to assess fear of water was developed and validated (Misimi et al. (2021)). Development and validity of the Fear of Water Assessment Questionnaire. Frontiers in Psychology). Therefore, I suggest that you include this information in the text (fourth or fifth paragraph in the introduction, i.e., somewhere between 69 and 93 lines) and comment on it and connect it to your idea.

·      Try to make the case for the need to develop PSPWC and its benefits to swim instructors and parents. You did this in the paragraph from line 68 to line 75. Try to modify this text to make it clear why it is important to develop this tool.

·      It is unusual to mention in the scientific paper how and how much you communicated with each other. I therefore suggest deleting the sentence in lines 107 to 109.

·       To use the same names of variables in Table 2 and in the text on page 7. I suggest using the names from the text.

·   To place the conclusion on content validity at the end of the first paragraph in discussion. You have correctly stated the aims of the study and the conclusion regarding the first aim, but the conclusion regarding the second aim is missing.

·       What is meant by study 1 (in line 374)? You have not mentioned this preliminary study so far. You should do so in the introduction or in the methods section.

·       What does [xxx] mean (in line 474)?

·  To be consistent with mentioning variables throughout the text. For example, in line 327, you used skill 7 (i.e., water entry by side). On the other hand, you wrote "water entry by side" (i.e., skill 7) in Conclusions.

-   To remove the sentences from line 579 to the end. Put them at the end of the Discussion section.

Author Response

Reply in the document
